# Older women's experience with COVID-19 pandemic: A study of risk perception and coping among culturally and linguistically diverse population in South Australia

**Noore Alam Siddiquee**[1], **Mohammad Hamiduzzaman**[2]*, **Helen McLaren**[3],
**Emi Patmisari**[3]

1 Graduate School of Public Policy, Nazarbayev University, Astana, Kazakhstan, 2 University Centre for Rural Health, School of Health Sciences, University of Sydney, Lismore, New South Wales, Australia, 3 College of Education, Psychology and Social Work, Flinders University, Adelaide, South Australia, Australia

☯ These authors contributed equally to this work.
* mohammad.hamiduzzaman@sydney.edu.au

**Data Availability Statement:** The primary data used for this study is accessible through the Harvard Dataverse platform, available for download at: https://doi.org/10.7910/DVN/OUGSUC.

## Abstract

### Background

A global catastrophe–the COVID-19 pandemic–appears to have two-dimensional health consequences for older adults: high risk of being infected and psychological distress. There is limited evidence on how the pandemic has impacted the life and coping of older adults who are culturally and linguistically diverse (CALD), women in particular. This study explored the COVID-19 risk perception and coping strategies of older CALD women in South Australia.

### Methods

A mixed-methods research design was employed, involving a 31-items coping and emergency preparation scale for survey and semi-structured interviews with participants. The older CALD women were approached through 11 multicultural NGOs. One hundred and nine women participants from 28 CALD communities completed the online surveys; 25 of them agreed to a telephone interview and provided their contact details. 15 older CALD women ultimately participated in interviews.

### Results

Mean sum-score of dread risk, unknown risk, and fear (M: 43.5; SD: 4.9) indicated that the participants were somewhat anxious and worried. Mean sum-score of coping (M: 79.8; SD: 9.3) reported their compliance with expert advice and disinfection practices but accessing health information (M: 2.8; SD 1.4) and tendency to minimize anxiety (M: 2.1; SD: 1.2) were below neutral. Significant variations were found in coping in terms of age, meaning that the women aged 75 years and older were less likely to cope with the pandemic ($P = 0.01$). Emergency preparation differed based on the participants' residence and occupation status.

**Funding:** The author(s) disclosed receipt of the following research grant for this project: Research was financed by the Flinders University COVID-19 Research Grant [Grant Number: 01.455.10977 - 2020] - Fund received - NAS, MH, HM. Funder website: https://www.flinders.edu.au/ The funders had no role in study design, data collection and analysis, decision to publish, or preparation of the manuscript.

**Competing interests:** The authors have declared that no competing interests exist.

The deductive-inductive thematic analysis of interview data was framed around three priori themes: risks of being affected, emotional and behavioral coping, and emergency preparation and access to services.

## Conclusions

Evidence shows a fear among the older CALD women with an endeavor to cope and prepare for emergency situations. This suggests the requirements for interventions that improve coping and reduce the risk of stress among them.

## Introduction

Since its outbreak the novel coronavirus-19 (COVID-19) has remained one of the major challenges of our time. This global catastrophe appears to have had two-dimensional health consequences for older adults: high risk of being infected and psychological distress. Besides its immediate and dire consequences on health and wellbeing, COVID-19 caused massive disruptions and uncertainties prompting policy measures and responses on a scale not seen before. Confronted with unprecedented crisis, governments around the world adopted and enforced numerous measures in seeking to contain and slow-down the spread of disease and its consequences. These measures ranged from safety protocols like hygiene and social distancing, to various forms of restrictions on social gatherings, and to enforcement of lockdowns. Other measures included travel restrictions, border closures and enforcement of quarantines and isolations. Although the measures were generally viewed positively in that they helped protect people from the COVID-19, they produced many undesirable effects at individual and societal levels [1]. Aside from causing possible negative emotions like depression, anxiety, stress and feeling of insecurity and helplessness [2,3], such measures left many people socially isolated and with reduced access to health and care services [4].

Though people of all ages were at risk of contracting COVID-19, older adults were particularly vulnerable and suffered severe consequences including high mortality rates [3,5]. This helps explain why several studies took interest in and focused on assessing the impact of COVID-19 on the elderly [6–8]. Recently published studies reported that older adults, especially those with chronic disease such as hypertension, cardiovascular disease, or diabetes, faced a particularly high risk of mortality due to COVID-19 [4,9]. Empirical studies found that while the risk perception of getting infected by COVID-19 tended to decrease as age increases, perceived severity was higher among the older people [10]. Other evaluations showed the devastating effects of the COVID-19 on the mental health of older adults [11–13]. In short, recent studies revealed how COVID-19 compounded health contexts for older people, many of whom already suffered from physical and mental health issues.

Older adults make up a significant proportion of Australia's population. Many of them were born overseas and speak languages other than English. Members of this group, commonly known as the culturally and linguistically diverse (CALD) population, are among some of the most disadvantaged in society. It must be noted that the word CALD lacks a universally accepted definition. In Australia, the term is generally used to refer to individuals who have cultural background different from the majority of Anglo-Celtic Australian culture. While some scholars tend to use other characteristics in defining CALD population [14], the term used in this paper refers to people who were born in non-English speaking countries and who do not speak English as the main language at home. They suffer from a range of adversity

associated with socioeconomic conditions, poor health and educational qualifications, lower employment levels and higher poverty [15], which were exacerbated during the COVID-19 pandemic with possible effects on their health and general welfare. Statistics show that older adults aged ≥60 years accounted for 90% of Australia's COVID-19 deaths. Furthermore, older CALD adults have a greater chance of being infected than non-CALD, irrespective whether they were first- or second-generation immigrants. The COVID-19 deaths for older CALD adults are 3 times higher than non-CALD (6.8 deaths per 100,000 compared to 2.3 deaths) [16]. The grey literature frequently reports a high prevalence of pre-existing chronic conditions and mental disorders (e.g., isolation, fear, anxiety, and stress) among the older CALD adults in Australia.

Research attention has rarely focused on understanding the situation of the CALD communities during the pandemic, and more specifically older people who may be some of the most vulnerable amongst them. In one of the very few studies, Mude et al. assessed the impacts of COVID-19 on CALD populations living in the greater Western Sydney region in Australia [17]. They found that the pandemic further complicated the situation of CALD populations; it caused major disruptions to employment, housing, relationships with partners, among others. Hamiduzzaman et al. explored risk perceptions and coping strategies amongst CALD adults in South Australia during the pandemic [11,12]. They found that COVID-19 caused considerable psychological and emotional burdens among CALD adults, due to their lack of health education and isolation owing to old age. While these findings are valuable, further disaggregation of data (e.g., by gender or age groups) is critical for informing specific interventions. Existing data showed that low-income women are 4 times more likely to die from COVID-19 than high-income men, and nearly 5 times more likely than high-income women [11,12]. It is, therefore, assumed that the low-income older women with CALD backgrounds may have suffered intersecting marginalization and compounding disadvantage during COVID-19, since they lack equitable access to information and services, have poorer health literacy, and are often underprepared for such disasters [18–20]. Hardly any attempt has been made to investigate and understand the effects that COVID-19 had on these women and their coping strategies. Their cognitive, emotional, and behavioral changes during COVID-19 continue to remain under-investigated. Consequently, little is known about the psychological impacts of the pandemic on older CALD women and their responses and preparations, which is important for future disaster or crisis planning.

This article explores the perspectives of older CALD women's risk perceptions and coping behaviors with the pandemic, adopting 'older' as over 60 years of age as defined by the United Nations [21]. This is a significant topic in the context of the Australian government's multicultural access and equity policy which seeks to ensure access and equity for all CALD people. Focused in the state of South Australia, COVID-19 impacted the economy and socio-cultural life in general, its healthcare and social services. South Australia saw three highly restrictive lockdowns, restrictions on international and domestic cross-border travels, and changes to business and governmental operations. This had the potential to impact older CALD women, many of them had family networks dispersed across Australia and internationally. This project builds on the collaboration between the South Australia's Multicultural Non-Government Organizations, Rural and Remote Health in South Australia, and Flinders University.

## Materials and methods

This mixed methods study involving survey and interviews received approval of the Flinders University Human Research Ethics Committee (Project ID: 2215) and complied with protocols compliant with the Declaration of Helsinki 2000.

## Theoretical assumptions

The risk perceptions and emotional and behavioural coping of older CALD women were investigated using the transactional model of stress and coping developed by Lazarus & Folkman [22]. This model addresses two aspects: (i) challenges and threats for an individual from her/his cognitive judgements of the meaning of a critical event; and (ii) the individual's capacity to respond to such events. The judgement of a critical situation depends on an individual's compassion or stress levels. Whereas the benign situations are judged as requiring no instrumental action, stressful situations are those that require specific actions. Stressful situations themselves fall into two categories, they can be either challenging or threatening. Challenging situations are those perceived to offer the potential for growth, mastery, and gain (e.g., performing well at exams). Threatening situations are those perceived to potentially result in harm or loss (e.g., performing poorly at exams). The perception of challenge or threat is determined in a secondary appraisal of one's ability to cope with, and respond to, the stressful situation. Challenge results from the judgment that one has the necessary resources to cope, and threat from the judgment that one does not. Although the implication of primary appraisal is that judgments of relevance or importance of a situation precede judgments of coping, this need not be the case. Secondary appraisals may, for instance, determine the initial relevance of a situation [22].

We conceptualized the risk perception as older CALD women's subjective judgement about the likelihood of being infected and affected by COVID-19. We also use Slovic's psychometric concepts [23]: cognitive (i.e., likelihood of being affected); affective (i.e., fear and general concerns); and psychometric (i.e., severity, controllability, and personal impact) to understand the women's risk perceptions. In theorizing coping with COVID-19 pandemic, the cognitive and behavioural aspects described by Lazarus & Folkman were considered [22], in that coping is constructed by how people adapt when dealing with or controlling stressful circumstances. It involves both emotional and behavioural precautions. Emotional coping was theorized in this study, using the microanalytic trait-oriented coping theory that indicates a bipolar dimension —repression and sensitization—where an individual copes with stress either by denying the existence or reacting with rumination and obsession. Emotional precaution refers to the cognitive efforts that can help to decrease the burden of a traumatic situation [22]. Some forms of emotional coping include staying busy, seeking social support, and having a positive mindset. Behavioral coping was understood as the use of protective measures (e.g., knowledge, trust in governments' initiatives, professional advice, restriction in movements) and emergency preparations (buying excess groceries, emergency plan, emergency contacts) [7,22].

## Research design

This paper was based on data gathered in a mixed-methods study conducted of older people in selected South Australian CALD communities, focusing on their risk perceptions, coping behaviors, and emergency preparedness during the pandemic. Specifically, the current study focused on a sample of older CALD women. Two major methods of data gathering were employed: an online survey and in-depth telephone interviews. Survey and interview data were collected concurrently from July to December 2020.

Multicultural non-government organizations (NGOs) working with CALD population were identified through the South Australian Directory of Community Services and approached to voluntarily distribute the Participation Information Sheet and a survey link to their community members. Of them, 11 multicultural NGOs agreed to support the project. To increase the number of participants, we used diverse strategies including the distribution of flyers, regular contacts with the NGOs, Facebook invitations, participant information

brochure distribution, displays at websites, and presentations to CALD community groups. The older women were recruited from self-nominated CALD communities, who lived in South Australia, and were 60 years and older [24]. Older women were discouraged to participate in this study if they were not from CALD background, and if they were diagnosed with any psychological conditions. 109 women participants from 28 CALD communities completed the online surveys; 25 of them agreed to a telephone interview and provided their contact details. 15 older CALD women ultimately participated in interviews (the remaining 10 were either unavailable or could not be reached). While survey participation was based on implied consent, written confirmation was obtained from each participant for their participation in interviews. The language used for both survey and interviews was English. This was made possible by the fact that the participants have had some level of English proficiency, even though their spoken language at home was not English.

## Online Survey: Variables, measures, and analysis

The 15-indicators risk perception and 31-items coping scales were used. All items had 5-point Likert scales (1 = "strongly disagree" to 5 = "strongly agree"). The risk perception scale was specific to COVID-19 risk perceptions, drawn from Gerhold's COVID-19 risk perception measures: cognitive, affective, and psychometric [25]. In brief, the risk perception scale includes 3-items asking respondents to rate the perceived future likelihood of becoming infected by COVID-19, and 12-items designed to elicit information about their personal feelings of dread risks, unknown risks, and fear of COVID-19. The coping with disasters scale has two sections; (i) emotional and behavioural precaution, and (ii) emergency preparedness, constructed based on Lazarus & Folkman's coping strategies [22]. The scale includes 23 emotional and behavioural strategies and eight emergency preparation ways.

The demographics were used as explanatory variables. Demographic data included ethnicity European, Asian, African, & South American); age groups (classified as 60–64, 65–69, 70–74, 75–79, 80–84, and 85+); residence (according to Modified Monash Model: large metropolitan areas, large rural towns, and small rural towns); education (no formal education, primary school, high school, Bachelor degree, or Masters and above), and occupation (formal vs informal labor force).

In the dataset comprising 109 participants, the surveys scales were missing for 6.42% (n = 7) participants. To address this, mode imputation was applied to fill the missing cases, with a separate imputation for each indicator. Following imputation, outliers were checked, and assumptions for parametric analysis were verified in the data. Descriptive statistics, such as response frequencies, means, and standard deviations, were calculated for presenting detailed demographics, risk perceptions of COVID-19, and coping with disasters. Demographics were regrouped for inferential statistical tests, such as location (metropolitan vs rural), ethnicity (European CALD vs other CALD groups), age (60–74 years vs 75≥ years), and education (≤ 11 years vs > 11 years). Independent-sample t-test was used (as the independent variables had two categories) to compare coping and emergency preparation sum-scores across demographics. Sampling weights have not been calculated in the analysis as no sampling method was used for data collection. All statistical analyses were done using SPSS (IBM version 23.0) and statistical significance was set at $p \leq 0.05$.

## In-depth interviews: Data and analysis

To gain further insights and a more comprehensive understanding of complex realities and nuanced lived experience of how these older CALD women tackled COVID-19, we held in-depth interviews with selected respondents. Individual telephone interviews were

approximately one-hour in duration conducted during November 2020-March 2021 with 15 women who had self-nominated to calls for interview participants or via the completion of their survey. The interview questions explored their emotional and behavioral precautions and emergency preparedness during the pandemic. Interviews were audio recorded and transcribed verbatim.

A deductive-inductive thematic analysis of data was conducted, guided by the method of Feredy and Muir-Chochrane [26] and involved the use of a priori-template with three preconceived themes to guide initial searching of the transcripts and sorting representative passages into each of the themes. Inductive coding of data within each of the three themes subsequently enabled sub-themes to be identified and the most meaningful of statements depictive of lived experience to emerge. The deductive-inductive process involved data immersion, reading and re-reading passages to get a deep sense of the core meanings from the perspectives of participants. Each step of the analysis was undertaken manually by two researchers (HM, EP) and conflicts resolved by a third researcher (MH). Representative statements from each of the theme were extracted for narrating and mixing.

## Results

Table 1 presents descriptive statistics of the demographics of the participants. A total of 109 older women completed the surveys and, in relation to ethnicity, 67% (n = 73) of them were European CALD, 27.5% (n = 30) were Asian, and rest of them were African (n = 4) and South

**Table 1. Descriptive statistics of demographics and the indicators of risk perceptions and precautions (n = 109).**

| Variables | Percentage (%) | Number (N = 109) |
|---|---|---|
| CALD Ethnicity | | |
| European-CALD | 67.0 | 73 |
| Asian | 27.5 | 30 |
| African | 3.7 | 4 |
| South American | 1.8 | 2 |
| Age group | | |
| 60–64 | 6.4 | 7 |
| 65–69 | 12.8 | 14 |
| 70–74 | 15.6 | 17 |
| 75–79 | 32.1 | 35 |
| 80–84 | 22.9 | 25 |
| 85+ | 10.1 | 11 |
| Area classifications | | |
| MM1 (Metropolitan) | 63.3 | 69 |
| MM3 (Large rural towns) | 0.9 | 1 |
| MM5 (Small rural towns) | 35.8 | 39 |
| Educational qualifications | | |
| No formal education | 16.5 | 18 |
| Primary school | 33.0 | 36 |
| High school | 22.0 | 24 |
| Bachelors | 16.5 | 18 |
| Masters and above | 11.9 | 13 |
| Occupation | | |
| Not in formal labor force (Pensioner, Housewife) | 92.7 | 101 |
| In Formal labor force | 7.3 | 8 |

American (n = 2) by background. In terms of age group, 32.1% (n = 35) were in the 75–79 years bracket and the second largest group was in 80–84 years (22.9%; n = 25). According to Modified Monash Model, most of the participants (63.3%; n = 69) in this study were from metropolitan areas (MM1), but a significant number of participants were found living in small rural towns (MM5–35.8%; n = 39). About 50.4% (n = 55) of the women had education level of high school or above, and 16.5% were with no formal education.

Table 2 illustrates the participants' risk perceptions, coping behaviors, and emergency preparation. The mean sum-score of the three items about infection risk was 7.9 (SD: 2.8), indicating a below neutral chance of being infected among the women. However, the mean sum-score of dread risk, unknown risk, and fear (M: 43.5; SD: 4.9) reported a somewhat agreement among the participants that they had feelings of anxiety, concerns, and fear because of COVID-19. The first item of 'dread, unknown risks and fear of being affected' had the highest mean score of 4.4 (SD: 0.9), which indicated that the women strongly identified the COVID-19 as a global disaster. These participants also strongly agreed (M: 4.3; SD: 0.8) with the statement that COVID-19 is something completely new to them. Many of them were not sure that they will not be affected by COVID-19 (M: 2.5; SD: 1.1).

Mean sum-score of coping was 79.8 (SD: 9.3), indicating agreement with the four items of coping behaviors [I listen to the experts and follow their advice (M: 4.0; SD: 0.9); I try not to do anything in rash (M: 4.0; SD: 0.6); I talk to family and friends to learn more about the situation (M: 4.2; SD: 0.6); and I wash and disinfect my hands more often than usual (M: 4.4; SD: 0.7). Interestingly, mean scores for accessing online health information (M: 2.8; SD 1.4) and tendency to minimize anxiety by eating, drinking, smoking, or taking medication (M: 2.1; SD: 1.2) were below neutral. Mean sum-score of emergency preparation was 23.9 (SD: 6.8) indicating a neutral position with the 8 indicators of emergency preparation. The participants somewhat agreed that they don't want to go shopping everyday (m: 3.6; SD: 0.9), but their response was below neutral when they were asked about collecting contact details for emergency services (M: 2.9; SD 1.0).

Table 3 confirms that the women's coping score significantly differed between age groups (60–74 years vs 75 years and over), but no variation was observed based on the participants' residence, ethnicity, education, and occupation. The emergency preparation was differed based on the participants' residence (metropolitan vs rural towns) and occupation status (not in labor force vs in labor force); while no significant variation was found based on the participants' ethnicity, age, and education.

While the quantitative analysis provided us with understanding of risk perceptions of COVID-19 and coping with back-to-back disasters, our thematic analysis offered insight into what the women may have perceived to be most at risk (e.g., people, health, or family) during a disaster and what they perceived were responsible for and/or required to ameliorate these risks (e.g., emotional coping, behavioral changes, information, or services). The analysis allowed us to observe patterns in the meaning of women's experiences, and to locate specific examples to elucidate further their perspectives of coping with disasters. Three priori themes that guided deductive coding and clustering codes into themes, thereby enabling the richness of participant experiences within each of following three themes.

## Theme 1: Risks of being affected

The risk of being infected and affected by the COVID-19 was shared by all participants. The infection risk was identified as high for older adults especially those with pre-existing medical conditions and co-morbidities, and the participants typically related their risk with various degrees of severity or even deaths. For example:

**Table 2. The participants' risk perceptions and coping behaviors (n = 109).**

| | Mean (SD) | Range (Min-Max) |
|---|---|---|
| **Risk of being infected (3 items)** | | |
| I might become infected by COVID-19 | 2.48 (1.1) | (1–5) |
| My family members might become infected by COVID-19 | 2.76 (0.9) | (1–5) |
| My friends might become infected by COVID-19 | 2.61 (1.0) | (1–5) |
| Sum-score of becoming infected | 7.93 (2.8) | (3–14) |
| **Dread, unknow risks and fear of being affected (12 items)** | | |
| COVID-19 is a global disaster | 4.4 (0.9) | (1–5) |
| COVID-19 will become more dangerous over time | 3.8 (0.9) | (1–5) |
| COVID-19 will affect future generations | 3.7 (0.8) | (1–5) |
| I can easily reduce the risk of infection | 3.5 (0.8) | (1–5) |
| The consequences of COVID-19 for me are my responsibility | 3.7 (0.8) | (1–5) |
| COVID-19 affects me personally | 3.2 (1.1) | (1–5) |
| COVID-19 is something completely new to me | 4.3 (0.8) | (1–5) |
| The effects of COVID-19 can be managed well | 3.4 (0.8) | (1–5) |
| The experts know about COVID-19 | 3.4 (1.2) | (1–5) |
| I know that I will not be affected by COVID-19 | 2.5 (1.1) | (1–5) |
| COVID-19 worries me | 3.7 (0.9) | (1–5) |
| I am afraid of being affected by COVID-19 | 3.6 (1.0) | (1–5) |
| Sum-score of anxiety, concerns, and fear | 43.5 (4.9) | (26–60) |
| **Coping behaviors (23 items)** | | |
| I listen to the experts and follow their advice | 4.0 (0.9) | (1–5) |
| I access online health information for awareness | 2.8 (1.4) | (1–5) |
| I think carefully about what to do and make informed plans | 3.8 (0.7) | (1–5) |
| I try not to do anything rash | 4.0 (0.6) | (1–5) |
| I talk to professionals who know about it | 3.2 (1.1) | (1–5) |
| I talk to family and friends to learn more about the situation | 4.2 (0.6) | (1–5) |
| I change things in my life to be able to cope better | 3.9 (0.7) | (1–5) |
| I've been thinking about what I usually do during other previous disasters | 3.2 (1.0) | (1–5) |
| I'm doing something completely new to cope with the circumstances | 3.7 (0.9) | (1–5) |
| I focus on my work or other activities to distract myself | 3.7 (0.8) | (1–5) |
| I imagine positive things to improve my mood | 3.8 (0.9) | (1–5) |
| I submit to my fate | 3.2 (1.1) | (1–5) |
| I tell myself things that make it easier for me to cope | 3.4 (0.9) | (1–5) |
| I wish I could change my worries and feelings | 3.3 (1.0) | (1–5) |
| I hope for a miracle | 3.2 (1.2) | (1–5) |
| I try to make myself feel better by eating, drinking, smoking, or taking medication | 2.1 (1.2) | (1–5) |
| I imagine times when it was better than today | 3.5 (1.0) | (1–5) |
| I try to leave the whole thing behind and want to rest or go on holiday | 3.2 (1.1) | (1–5) |
| I refuse to believe what is happening | 2.2 (1.1) | (1–5) |
| I wash and disinfect my hands more often than usual | 4.4 (0.7) | (1–5) |
| I avoid public places/events | 3.8 (0.8) | (1–5) |
| I avoid public transports (tram, bus, train) | 3.7 (0.9) | (1–5) |
| I avoid contact with risk groups (old people and people with previous/current illnesses) | 3.5 (1.0) | (1–5) |
| Sum-score of coping behaviors | 79.8 (9.3) | (47–104) |
| **Emergency preparedness (8 items)** | | |
| I bought more food than usual | 2.9 (1.1) | (1–5) |

(*Continued*)

**Table 2.** (Continued)

| | Mean (SD) | Range (Min-Max) |
|---|---|---|
| I bought larger quantities of hand disinfectant/soap | 3.3 (1.2) | (1–5) |
| I bought larger amounts of staple foods (flour, sugar, pasta, rice, canned food) | 3.0 (1.2) | (1–5) |
| I bought large quantities of toilet paper and other hygiene items | 2.8 (1.3) | (1–5) |
| I don't want to go shopping every day | 3.6 (0.9) | (1–5) |
| I buy large quantities of special offers | 2.9 (1.0) | (1–5) |
| I deliberately store essential goods in order to be prepared | 2.5 (1.0) | (1–5) |
| I collect all emergency services contact details | 2.9 (1.0) | (1–5) |
| Sum-score of emergency preparedness | 23.9 (6.8) | (8–37) |

**Notes:** Max indicates Maxim4um; Min indicates Minimum; SD indicates Standard Deviation.

*A virus that will attack your lungs especially for our age groups; more vulnerable if one has pre-existing medical conditions; community transmission if no precautions like wearing of mask are not taken.* (Anita, 69 years old)

*The virus is highly contagious. And the people that are at risk are those that are over 70. . . . if you caught the virus, it could present itself in various degrees of severity ranging from just a slight cold to the inability to breathe and having to be on a ventilator.* (Gillian, 72 years old)

*Causes serious illness in older people, and potential death.* (Kathryn, 62 years Old)

One woman described the COVID-19 as more serious than the previous epidemics such as Spanish flu or bubonic plague:

*. . . more serious than Spanish flu and periodic bubonic plague and all sorts of things of the Middle Ages; it is highly contagious.* (Rebecca, 78 years old)

**Table 3. Participants' characteristics and distribution of coping and emergency preparation scores across different categories (N = 109).**

| Variables | Coping | | t-value | p-value | Emergency preparation | | t-value | p-value |
|---|---|---|---|---|---|---|---|---|
| | Mean | SD | | | Mean | SD | | |
| **Location** | | | 0.91 | 0.36 | | | 2.57 | **0.01** |
| Metropolitan, n = 69 | 80.23 | 8.5 | | | 23.03 | 6.5 | | |
| Rural towns, n = 40 | 81.93 | 10.7 | | | 26.23 | 5.8 | | |
| **CALD ethnicity** | | | 0.72 | 0.47 | | | 0.25 | 0.80 |
| European CALD, n = 73 | 78.74 | 9.2 | | | 24.08 | 6.5 | | |
| Other CALD Groups (Asian, African & South American), n = 36 | 80.11 | 9.5 | | | 23.74 | 7.2 | | |
| **Age (in years)** | | | 2.52 | **0.01** | | | 0.44 | 0.66 |
| 60–74 years, n = 38 | 82.19 | 5.7 | | | 23.40 | 3.2 | | |
| 75 years and over, n = 71 | 78.60 | 7.7 | | | 22.82 | 7.8 | | |
| **Education level** | | | 1.67 | 0.90 | | | 1.79 | 0.07 |
| Less than 11 years of schooling, n = 78 | 78.51 | 8.7 | | | 24.75 | 7.2 | | |
| More than 11 years of schooling, n = 31 | 81.68 | 9.5 | | | 22.13 | 6.1 | | |
| **Occupation status** | | | 1.07 | 0.28 | | | 3.42 | **0.002** |
| Not in formal labor force, n = 101 | 80.06 | 9.4 | | | 24.20 | 7.01 | | |
| In formal labor force, n = 8 | 76.38 | 7.9 | | | 20.63 | 2.2 | | |

Interestingly, a few women portrayed understanding of the causes and transmission of the infections, symptoms, and potential health problems, for examples:

*The symptom is like flu and fever and the pain of the body and then it affects everything; it will spread so quickly.*

*It's a highly contagious virus that affects everybody, in particular the weaker people with weaker immune systems and other comorbidities; It can be caught quite easily. Airborne and on surfaces. And it works quickly to destabilize the lungs, but I think other areas of the body as well; some people are not taking it seriously.* (Prinaka, 68 years Old)

*Attacks your lungs like concrete setting in your lungs; develops pneumonia; have to go on life support especially the vulnerable people aged over 60.* (Selina, 66 years old)

Perceptions of health risk were amplified by the participants who shared their awareness that they, as older women, were at greater risk of being affected. For them the risk of being affected meant the economic impacts; given the lack of sufficient support from the government it also caused psychological anxiety amongst some of them. Two participants reported their concern about business and economy, with one claiming that the restaurant business run by her family suffered in dropping demands and profits, which affected her mental health. One of them was worried that there was no assurance of government support should COVID-19 lead to recessions.

In addition, some participants had a perception that the COVID-19 would not leave. Therefore, they were unsure about its long-term effects, as one woman said:

*It is extremely contagious, that we have lots of precautions; it's not gonna [going to] leave us; people getting respiratory failure, the people who are called the end with, particularly, with chronic conditions, people with diabetes, heart problems, and so on, can be extremely vulnerable. We don't quite know, the long-term effects of it; it needs to be treated and isolated.* (Zunu, 67 years old)

Two participants were not sure about the risks of being affected by COVID-19. This uncertainty was due mainly to a lack of accessibility of information about COVID-19. One said that she did not know of the risks of COVID-19 because nobody she knew talked about it. In other case the respondent was confused about the risks because there was too much information, especially on online platforms about the pandemic, and at times they are contradictory.

*Because the flood of information changes from day to day, it is very confusing. When the information is very confusing, it does not help.* (Selina, 66 years old)

## Theme 2: Emotional and behavioral coping

The coping with the pandemic was described by the participants in relation to emotional and behavioral precautions. The descriptions of emotional precautions started with the identification of the pandemic as a multi-faceted issue and coping with uncertainty around recovery from the disease. For examples:

*It has long-standing and life-threatening issues; it's not a flu; some people will have lots of ongoing issues. Some will make grandpa some will need ongoing heavy drugs help and will*

*never be able to work again. So, it's a very broad-spectrum issue that I don't think it has been advertised enough as dangerous as it is.* (Nina, 71 years old)

*Some people recover from it, and some people don't; it can damage lungs and all that sort of stuff. I am afraid of going out to shopping and visiting my 90-year-old mum.* (Wendy, 66 years old)

Emotional coping was found difficult because of the uncertainty of COVID-19 and isolation. However, they often drew emotional support and wellbeing from the family members and friends, as two participants described:

*That's right, for emotional support and the well-being we mostly rely on the family and friends.* (Rebecca, 78 years old)

*Even though I don't see a lot of my friends now, we still keep in touch on the phone. The only challenge, the only thing that I miss traveling and things like that. I mean, I like to go to Malaysia and elsewhere for holiday or whatever. But obviously, that can't be done at the moment.* (Jennene, 73 years old)

In terms of behavioral coping with the COVID-19 pandemic, most of the participants reported having used several measures including self-isolation, social distancing and avoiding social activities:

*I've self-isolated a lot. Anyway, I have a few co-morbidities. So, I'm a high-risk person anyway. I've done everything that's been required for the higher risk people.* (Rebecca, 78 years old)

*So back in March when it all started, or at least when we were made aware of the pandemic and were advised to go into isolation and the details came out more clearly. We certainly adhered to the thing at home and stopped, for example, the family gatherings that we used to have at my daughter's place on a weekly basis.* (Ruby, 61 years old)

*The most important thing to mitigate the risk is to obey the social distancing rule that has been mandated.* (Ellina, 65 years old)

*I don't hug people and I don't shake hands. And I became a Japanese. So, I will bow my head to people.* (Prinaka, 68 years Old)

Living in rural areas advantaged some participants in terms of self-isolation or social distancing, as one said below, but some of them did not comply with any of the measures.

*. . . we live on three acres. So, we've got plenty of rooms to move around.* (Nina, 71 years old)

*I was just free. I just go in and out. I don't go out much. You know, I go to the shop to get a few things—only at Coles.* (Zunu, 67 years old)

A few participants expressed their concerns because they found people not observing physical distance and were coughing and sneezing in public places.

*If you're over 70; one of the major risks is trying to say, going out in public places where people are not observing the distances apart, the 1.4 meters or the people are not respecting it at all. I also find people who cough and sneeze and stuff like that into the air without covering the mouth.* (Kathryn, 62 years Old)

The behavioral measures practiced by the participants also included hand sanitization, washing hands, and wearing masks, aside from getting vaccinated.

*We have a table at the front door with the hand sanitizer on it. Wherever I go I will wear face masks.* (Nina, 71 years old)

*For the first time. I got a flu vaccine, right. You should get flu vaccine. So, we had hand sanitizer in the car, which we don't normally have. So, we've been washing our hands and sanitizing more frequently.* (Gillian, 72 years old)

*I wear masks for wherever I am, and whatever I do. And I maintain hygiene by washing my hands and, and so forth.* (Ruby, 61 years old)

One participant did not respond about her coping strategies, given that she was unsure about the consequences of the pandemic. Another participant expressed her confusion about the information in the different websites, which had negative impact on her decisions about coping strategies and preparedness.

## Theme 3: Emergency preparation and access to services

The majority (11) of participants did not report any changes associated with emergency preparedness and access to and use of health information and social or health services during the pandemic. Four participants stated that they carried on with life as usual. Four did not think about preparing for COVID-19 while it was getting worse in Australia. Three indicated that they were following the recommended health precautions, and so they were prepared even though not having experienced emergency situations. However, some important changes were found in the participants' descriptions of their daily lives during the pandemic, including their plans, preparations, and usage of services.

Having an action plan was one of the infrequently described phenomena where few of the rural older CALD women felt that they could use the action plan during the pandemic they have from previous disasters. Although they did not disclose the key elements of the action plan, it could be assumed that this action plan was about emergency safety issues, critical information and contact details of local hospitals and GPs. As one participant stated:

*I live in a bushfire area, so, we have our action plan. We have briefcase with all our documentation. We always keep that next to us. And we have a lovely care system. Because we're 20 minutes from an ambulance service, either way, and we live 45–50 minutes from my GP clinic.* (Nina, 71 years old)

This was echoed by another participant who described the way she collected contact numbers of important people and organizations and detailed her skills in managing virtual communication during the pandemic, while many participants lacked such skills. For examples:

*I have collected contact numbers of various people from where I can get help, and I find an ambulance plan. So, I can definitely call the ambulance; I have various devices, I have access to data, and I can use my phone, my iPad, my computer. . . . I have skills to be able to do that* [managing emergency contacts] *either through zoom, WhatsApp, FaceTime, Skype, whatever the medium would be; haven't had an emergency so far.* (Kathryn, 62 years Old)

Some participants also reported changes in terms of their access to and utilization of healthcare services. Whilst many participants were hesitant to answering the question about the use

of social media for accessing healthcare, two participants used telehealth in their interactions with general practitioners, specialists, and pharmacists while complying with social distancing rules and mitigating fear associated with exposure to contagions. One of them said:

> *I did zoom calls to seek consultation with my GP and my specialist, my endocrinologist would give them calls and that we consulted that way. And then, for my discussion, I then registered with them. What do you call it? And I'll place order for medicines digitally and then pay for it with my credit card. And one of the staff members of the chemists delivered to my door outside when they finished work. And they let me know it's there so that I can collect it.* (Prinaka, 68 years Old)

Most of the participants shared their experiences of online shopping of groceries, as stated by two:

> *We tended to shop in smaller shops. So, we stopped shopping in places where there are lots of people busy shopping who were not aware of social distancing. . . . Instead, we started to shop online mostly except for food shopping.* (Gillian, 72 years old)
>
> *Everything was done online for the last two months.* (Kathryn, 62 years Old)
>
> *From the day one, we ordered online, and we get groceries delivered. . .* (Prinaka, 68 years Old)

In summary, while some older CALD women showed their competence in coping, most of them were found with no action plans and preparations for the consequences of the pandemic.

## Discussion and implications

Given the lack of studies focused on how various vulnerable groups managed during the COVID-19 pandemic, we aimed to investigate the experience of one of Australia's most disadvantaged groups—the older women in CALD communities with a specific focus on risk perceptions, coping behaviors among them and the strategies they employed during the pandemic. This is the first of its kind—to the best of our knowledge—where the experience of this specific group of people was investigated and documented. It showed current levels of risk perception among the older women in South Australia's CALD community and how they have reacted and managed during stressful situations during the COVID-19 pandemic.

In general, the findings show high levels of risk perception and behavioral coping among the older women in South Australia's CALD communities who participated in the study. The risk of being infected by COVID-19 showed as particularly high amongst those with pre-existing health issues. The findings were broadly consistent with other studies elsewhere that showed females to perceive COVID-19 related risks more than males and, consequently, were more active in following preventive and coping strategies than their male counterparts [27,28]. Interestingly, our study finds some variations between age groups: lower levels of coping and emergency preparation was reported by those in the oldest age group (75 years and above), compared to those in 60–74 years of age a group. This finding in our South Australian study aligns with similar studies elsewhere where age was found to be negatively associated with risk perceptions, and pre-cautions/preparations [29–31]. One possible explanation for such difference is that those individuals in the oldest of age groups may have experienced similar crises during their lifetime developing better resilience in living with adversities and challenges. Existing research suggests that older adults are more resilient than the younger adults, they have higher emotional regulation and skills to cope with the adversities [32].

The findings of our study reveal that the older women in the CALD community have coped with the pandemic reasonably well, by minimizing risk exposures and adopting precautions in terms of health and hygiene. As elsewhere, reducing risk of exposure to COVID-19 was a common coping strategy employed by them during COVID-19, where they followed health protocols and advice like staying at home, wearing face masks, avoiding crowded places and social activities, shaking hands/hugging/kissing, and frequently washing, cleaning, and sanitizing with disinfectants [33,34]. Not surprisingly, some older women reported shopping online and using telehealth/telemedicine and video conference services as a way of maintaining physical distancing and avoiding personal contact during lockdowns and restricted movements. While this gave them some degree of protection from being infected by COVID, and flexibility and convenience when accessing vital services, and avoiding hassles of transport, and other inconveniences, such changes were reported in a limited number of cases. Although it is difficult to judge how effective such arrangements were, the women's interview transcripts revealed that services were of important concern to them during the pandemic. Nevertheless, this reflects an evolving trend in service delivery which is likely to see expansions in future adding to efficiency and convenience on both sides involved.

The study also showed that older CALD women attempted emotional coping, as reflected in high scores in some strategies (e.g., imagining positive things to improve mood, thinking about times when it was better, and doing completely new things to cope with the circumstances). In short, we found that older women used both problem-focused as well as emotionally-focused strategies in coping with the pandemic situation [35]. They also followed expert advice like washing hands and avoiding contacts with risk groups, as common elsewhere. A point of departure in the South Australian context was that, unlike in other jurisdictions, the participants were not buying and stockpiling more food and other essential supplies than the usual. This could be attributed to the relatively less severe nature of the pandemic in South Australia at the time of the study and consequently limited nature of the lockdowns imposed which meant that there was less fear of total disruption of the supply chain.

A notable finding of the present study is that aside from health-related issues and concerns COVID-19 has caused considerable stress and anxiety among older CALD women who already suffer from a range of disadvantages. Even though South Australia is among the least affected Australian states and territories in terms the severity and the number of COVID-19 related hospitalization and deaths, the pandemic created an atmosphere of fear and anxiety among the older women, especially those in the CALD communities. It had significant impact on the life and wellbeing of this cohort of population. The isolation under COVID-related lockdowns and restrictions, while helped contain the outbreak, for the older people such measures have increased their isolation and marginalization. For many, isolation itself under COVID-19 related lockdowns were distressing. This was compounded further by reduced contacts and opportunities to visit and socialize with other community and family members. In consideration of the discursivity of family and nurturing responsibilities in women's lives [36], the importance women placed on family relations and support of others was likewise reflected in text mining results and thematic findings.

The economic fall out of the COVID-19 and associated anxiety was evident. One respondent revealed her anxiety by sharing that her family restaurant business experienced reduced demand and profit since the start of COVID-19. This caused worry about the well-being of her family, as an example of women's carriage of additional burden in the form of emotional labor during the pandemic [37]. In other cases, respondents reported increased experience of racism and discrimination based on culture and ethnicity, which contributed to additional social isolation and anxiety. Such findings were consistent with the current literature that shows that challenges associated with COVID-19 have led to increased mental and psychological distress

and that COVID-19 has acted as a catalyst to expose some of the disadvantages within the CALD community [17,38]. Thus, the present study underscores the importance of paying attention to an important but often overlooked aspect -the impact of COVID on emotion and mental health of older women, on top of pre-existing social issues that intersect and compound disadvantage of older women in CALD communities. As shown, although older CALD women faired reasonably well in terms of behavioral coping by adopting measures like self-isolation, social distancing and avoiding social events etc., emotional coping was found to be particularly hard and problematic. As a way of coping in this situation the older women in the CALD community relied heavily on family and friends as a source of emotional support during the difficult and challenging time.

The present study showed that, in pandemic situations like this, access to information was critical, especially for people with low levels of English proficiency and inadequate digital literacy. The study found that inadequate health messaging, together with low levels of English and digital literacy, served as a major impediment to pandemic preparedness and access to available services. It revealed that for some older women information about COVID was inadequate, for others it was too excessive and at times contradictory, which added to their confusions. This was perhaps due to different levels of education and digital literacy [39]. Those with fair amounts of such knowledge experienced a flood of information about COVID on digital platforms, coming from diverse sources with frequent changes [40,41]. Others who relied on more traditional sources may have found information rather limited and inadequate. In both cases, the constantly evolving nature of information and global communication led to confusions especially for those with limited English, as they relied on social media and informal sources for vital information. Some of those interviewed in the study lamented the lack of timely, accurate and adequate, and accessible COVID-19 information, up-dates, and advice in languages other than English for the benefit of the CALD population. This explained why there were people within the community who were not fully aware of the risks involved at that time. Thus, the findings of the study highlight the importance of addressing an obvious gap that exists in terms of communication between older adults in the CALD community, and the public health information dissemination systems and processes [42]. Needless to add that effective communication and access to relevant information about the pandemic could have led to a new level of coping and preparedness by promoting the adoption of problem-focused precautions.

Another notable finding was that although older women perceived COVID-19 as a serious threat and a global challenge, a challenge more serious than the Spanish flu, preparations to deal with the consequences of the pandemic were experienced as inadequate. In other words, the majority of those who participated in the study were un- or under-prepared in terms of their responses. Only a small number of those surveyed and interviewed were sufficiently ready in terms of action plans and contingencies. This suggests that there was work to be done in raising awareness and capacity of relevant groups to undertake appropriate courses of action, especially in the event such emergencies get out of hand requiring immediate and drastic measures than those that were followed.

## Conclusions

Older CALD women had high risk perceptions of acquiring COVID-19 and becoming sick from its effects. This high-risk perception prompted them to take strong preventative measures towards contracting COVID-19 and were therefore highly resilient during the crisis. This included washing hands and disinfecting surfaces regularly. The maternal nature of these older CALD women influenced their risk perceptions, placing significant concern towards the safety of their family, followed by their friends, and lastly, themselves. Despite the

unprecedented situation of a global pandemic, these women focused on what they could control and not on what they could not. They engaged themselves in positive tasks that kept their minds busy and inspired high morale. This included staying well informed about staying safe during the pandemic and keeping in touch with family and friends. Whilst older women fared well during the pandemic, better social support and less social isolation is a key consideration for the future, as socially isolated individuals did not fare well in comparison to their counterparts who remained surrounded by family. Older CALD women decreased exposure risk by limiting their trips to buy food and following government health advice.

This cross-sectional study provided a "snapshot" of the perceived risk, coping mechanisms, and emergency preparedness of older women from the CALD population. Recent research has concluded that web-surveys are feasible and a suitable alternative for surveying older people [43], however we acknowledge that by not offering paper-based or different language alternatives that this may bias results by excluding subsamples who may lack digital or English literacy. Data recruitment and collection methods were impacted by COVID-19 restrictions, limiting opportunity for distribution of materials via public meetings and snowballing, which will have implicated survey and interview response rates. Other limitations of this study, as in other cross-sectional studies in general, include difficulty in drawing causal conclusions. Identified associations may be difficult to interpret, unable to investigate relationships between outcomes and risk factors, and are also prone to bias. The sample in this study does not represent the entire CALD population in Australia proportionally. In future research it would be interesting to see a larger CALD populations of older women surveyed in additional locations within Australia on disaster issues affecting them. This would yield more meaningful data and allow responses to be compared between other states and territories. The findings of this research may help to determine which CALD communities fared better during natural disaster, helping to guide more effective policies, particularly in terms of preparedness.

## Supporting information

**S1 Checklist. Human participants research checklist.**
(DOCX)

**S1 File.**
(DOCX)

## Acknowledgments

We are thankful to the participants who provided time and shared their experiences about the COVID-19. We acknowledge the contribution of CALD community organisations that helped the researchers by encouraging people to participate in this study. We thank Judy Baily, Research Assistant at Flinders University Rural Health SA for her valuable support with recruitment of participants and data collection.

## Author Contributions

**Conceptualization:** Noore Alam Siddiquee, Mohammad Hamiduzzaman, Helen McLaren.

**Data curation:** Mohammad Hamiduzzaman, Helen McLaren.

**Formal analysis:** Noore Alam Siddiquee, Mohammad Hamiduzzaman, Emi Patmisari.

**Funding acquisition:** Noore Alam Siddiquee, Mohammad Hamiduzzaman, Helen McLaren.

**Investigation:** Noore Alam Siddiquee, Mohammad Hamiduzzaman, Helen McLaren.

**Methodology:** Noore Alam Siddiquee, Mohammad Hamiduzzaman, Helen McLaren, Emi Patmisari.

**Project administration:** Mohammad Hamiduzzaman, Helen McLaren.

**Resources:** Mohammad Hamiduzzaman.

**Software:** Mohammad Hamiduzzaman.

**Supervision:** Noore Alam Siddiquee, Mohammad Hamiduzzaman, Helen McLaren.

**Validation:** Noore Alam Siddiquee, Mohammad Hamiduzzaman, Helen McLaren.

**Visualization:** Mohammad Hamiduzzaman.

**Writing – original draft:** Noore Alam Siddiquee, Mohammad Hamiduzzaman, Helen McLaren, Emi Patmisari.

**Writing – review & editing:** Noore Alam Siddiquee, Mohammad Hamiduzzaman, Helen McLaren, Emi Patmisari.

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
