## [Decision Letter · Decision Letter 0]

13 Nov 2023

PONE-D-23-24279Manuscript: Older women experience with COVID-19 pandemic: A study of risk perception and coping among culturally and linguistically diverse population in South AustraliaPLOS ONE

Dear Dr. Hamiduzzaman,

Thank you for submitting your manuscript to PLOS ONE. After careful consideration, we feel that it has merit but does not fully meet PLOS ONE’s publication criteria as it currently stands. Therefore, we invite you to submit a revised version of the manuscript that addresses the points raised during the review process.

We look forward to receiving your revised manuscript.

Kind regards,

Sonu Bhaskar, MD PhD

Academic Editor

PLOS ONE

Journal Requirements:

4. Please include captions for your Supporting Information files at the end of your manuscript, and update any in-text citations to match accordingly. Please see our Supporting Information guidelines for more information:

Additional Editor Comments (if provided):

Thank you for submitting your work to PLOS One. Based on the careful review of your submission, and taking into account the feedback from two independent reviewers, we invite you to revise your work and provide point-by-point rebuttal to the comments.

Reviewers' comments:

Reviewer's Responses to Questions

**Comments to the Author**

1. Is the manuscript technically sound, and do the data support the conclusions?

Reviewer #1: Yes

Reviewer #2: Partly

2. Has the statistical analysis been performed appropriately and rigorously? 

Reviewer #1: Yes

Reviewer #2: I Don't Know

3. Have the authors made all data underlying the findings in their manuscript fully available?

Reviewer #1: No

Reviewer #2: Yes

4. Is the manuscript presented in an intelligible fashion and written in standard English?

Reviewer #1: Yes

Reviewer #2: Yes

5. Review Comments to the Author

Reviewer #1: The paper examines the issue of experiences of COVID-19 in CALD older women in South Australia, particularly their risk perception and coping strategies. This is an important study. However, there are significant issues the authors would need to address before this paper can be published. The major concerns are around:

1. Handling of missing data. how many participants were dropped? How the missing data for the remaining participants were handled? It would also be important to highlight more clearly about this, given the small sample size in this study.

2. Analysis of outcome variables. How did the author analyse the outcome variables? Did they score the mean or sum the scores for each item? If so, supported with evidence from the literature, are there baselines or thresholds that indicate negative or positive experiences?

3. Did the authors assess the data to ensure it met the assumptions for the independent sample t-test? Further information is required.

4. Qualitative analysis. Who analysed the qualitative data? How many people were involved in the analysis? How were divergences resolved if two or more people were involved in the analysis? How long did the interview last? Where were the interviews done? Were these recorded? transcribed? If so, how were they done? Did the author use any software program to help with this qualitative analysis?

General considerations around the following points also need to be considered. The paper requires careful revision for:

1.grammar

2. use of language. For example, “native Australians” is rarely used to describe Australia’s First Nations. Consider using Australia’s First Nations, Aboriginal and Torres Strait Islanders, or Indigenous Australians.

I have also made some comments on the paper for you to consider

Reviewer #2: The methodology section should be more elaborate.

Time of the survey should be mentioned precisely.

Manuscript is written in standard English.

I am not a statistician so i am unable to comment on statistical analysis part.

6. PLOS authors have the option to publish the peer review history of their article (what does this mean?). If published, this will include your full peer review and any attached files.

Reviewer #1: No

Reviewer #2: No

---

## [Author Response · Author response to Decision Letter 0]

19 Nov 2023

17 November 2023

To

Editor

PLOS ONE

RE: Ms. No. PONE-D-23-24279 - Older women experience with COVID-19 pandemic: A study of risk perception and coping among culturally and linguistically diverse population in South

Australia. 

Dear Editor,

Thank you for the opportunity to strengthen our manuscript submission. We found the review comments positive and helpful and have addressed them in the point-by-point table below. A point-by-point response letter has been accompanied with our revised manuscript with track changes and clean version of the final manuscript. 

This letter provides a detailed response to each reviewer point raised, describing exactly what amendments have been made to the manuscript text and where these can be viewed. All changes to the manuscript are indicated in the text by highlighting or using track changes. We thank you for your time and look forward to hearing from you regarding our manuscript. 

Reviewer 1

The paper examines the issue of experiences of COVID-19 in CALD older women in South Australia, particularly their risk perception and coping strategies. This is an important study. However, there are significant issues the authors would need to address before this paper can be published. The major concerns are around:

Handling of missing data. how many participants were dropped? How the missing data for the remaining participants were handled? It would also be important to highlight more clearly about this, given the small sample size in this study. The survey and analysis section has been reviewed as follows: 

In the dataset comprising 109 participants, the surveys scales were missing for 6.42% (n=7) participants. To address this, mode imputation was applied to fill the missing cases, with a separate imputation for each indicator.

1. Analysis of outcome variables. How did the author analyse the outcome variables? Did they score the mean or sum the scores for each item? If so, supported with evidence from the literature, are there baselines or thresholds that indicate negative or positive experiences? This survey and analysis section has been reviewed to avoid confusion.

Mean-sum-scores were used in the analysis: 

Mean sum-score of coping was 79.8 (SD: 9.3), indicating agreement with the four items of coping behaviors. 

And 

Mean sum-score of emergency preparation was 23.9 (SD: 6.8) indicating a neutral position with the 8 indicators of emergency preparation. 

Our study focused on comparing the participants’ agreements with coping and emergency preparation across their demographics. 

2. Did the authors assess the data to ensure it met the assumptions for the independent sample t-test? Further information is required.

 The following information has been added to the statistical analysis section to address this review comment: 

Outliers were checked, and assumptions for parametric analysis were verified in the data.

 …

Independent-sample t-test was used (as the independent variables had two categories) to compare coping and preparation scores across demographics

3. Qualitative analysis. As requested, we have added detail as appropriate. 

a. Who analysed the qualitative data? This has been added, “HM, EP, conflicts resolved by MH”. 

b. How many people were involved in the analysis? Three is obvious based on the previous response. 

c. How were divergences resolved if two or more people were involved in the analysis? Resolved by a third person, as in previous comments. 

d. How long did the interview last? While much of this is stated in Methods/Research design, we have also added to the to the Methods/In-depth Interviews: data and analysis subsection, “Individual telephone interviews were approximately one-hour in duration conducted during November 2020-March 2021 with 15 women who had self-nominated to calls for interview participants or upon via the completion of their survey.“

e. Where were the interviews done? 

f. Were these recorded? transcribed? If so, how were they done? We originally stated, “Interviews were audio recorded and transcribed verbatim.”

g. Did the author use any software program to help with this qualitative analysis? We added, “Each step of the analysis was undertaken manually by two researchers (HM, EP) and conflicts resolved by a third researcher (MH).”

General considerations around the following points also need to be considered. The paper requires careful revision for:

1. grammar A native English-speaking author has corrected grammar throughout.

2. use of language. For example, “native Australians” is rarely used to describe Australia’s First Nations. Consider using Australia’s First Nations, Aboriginal and Torres Strait Islanders, or Indigenous Australians. This has been corrected to “native born Australians”. We have also added to the CALD comparison description “irrespective whether they are first- or second-generation immigrants” to make this clear that we are not referring to Australia’s first Nations peoples. 

Reviewer 2

1. I have also made some comments on the paper for you to consider. Thank you, we have worked thought these comments and responded as requested by this reviewer.

Comment from the paper re “The coping with disasters scale has two sections; (i) emotional and behavioural precaution, and (ii) emergency preparedness, constructed based on Lazarus & Folkman’s (1984) coping strategies. The scale

includes 23 emotional and behavioural strategies and eight emergency preparation ways. In addition to 31 separate indicators, two sum-scores (23 items of coping and 8 items of emergency preparation) were considered.”

a. How did the author analyse the outcome variables? Did they score the mean or sum the scores for each of the items? If so, is there a baseline or threshold which indicates negative or positive experience? This section has been reviewed to avoid confusion.

Mean-sum-scores were used in the analysis: 

Mean sum-score of coping was 79.8 (SD: 9.3), indicating agreement with the four items of coping behaviors. 

And 

Mean sum-score of emergency preparation was 23.9 (SD: 6.8) indicating a neutral position with the 8 indicators of emergency preparation. 

Our study focused on comparing the participants’ agreements with coping and emergency preparation across their demographics. 

Comment from the paper re – “Independent-sample t-test was used to compare coping and preparation scores across demographics. Sampling “

b. How did the authors score the measured items? Did the authors assess the data to ensure it met the assumptions for independent sample t-test? Further information is required The following information has been added to the statistical analysis section to address this review comment: 

Outliers were checked, and assumptions for parametric analysis were verified in the data.

 …

Independent-sample t-test was used (as the independent variables had two categories) to compare coping and preparation scores across demographics

Comment from the paper re – “the theme were extracted for narrating.”

c. How was this done? did the author use any software program to help with this qualitative analysis? How many people were involved in the thematic analysis? Who were the authors? If more than two people were involved in the analysis, how were divergences in the analysis resolved?” This has been revised per Reviewer #1 above (it is not clear whether this is the same reviewer responding on the paper as in comments by reviewer #1).

Comment from the paper re – “Not surprisingly, 92.7% participants were not involved in the formal labor force.”

d. Not intuitive, especially when participants are older. consider removing. Sentence removed. 

Comment from the paper re age group 60-64 in Table 1

e. This include people who are not retired. Consider including retirees only. It may be accurate to describe people in the working age group as 'older' to avoid running the risk of ageism We have specified in response to an earlier comment re ‘older’ and clarified our use of >60 years based on UN age definition. We are not revising the age groups in our sample considering that there is no retirement age for women in Australia unless a member of judiciary (pension eligibility age for women is 67), and we cannot identify retirees in our sample.

We have also corrected an erratum in which we referred to ‘Elderly’ – changed to consistent use of ‘older’ considering the UN definition as stated. 

Comment from paper re quotes.

f. Are these quotes taken from one participant? consider attributing quotes to participants. Pseudonym names and age of the participants have been added in brackets after each quote

Comment from paper re qualitative lead-in.

g. How many few is few? In qualitative research, the focus is on the richness and depth of information. consider using other terms. for example instead of saying few say three people or two people etc. As a team of researchers, we have members skilled in qualitative analysis who disagree. Since the emphasis is on richness and depth, numbers may be used. However it is common for qualitative reporting to use terms such as ‘few’, ‘some’, ‘many’, etc.

2. The methodology section should be more elaborate.

 We have elaborated on the methodology, as in our feedback to Reviewer #1. We have also commenced the methods section with providing research ethics approval and ethics convention. 

3. Time of the survey should be mentioned precisely.

 We have added in Methods/Research design, “Survey and interview data was collected concurrently from July to December 2020.”

4. Manuscript is written in standard English. Yes

5. I am not a statistician, so I am unable to comment on statistical analysis part. We have checked the statistical analysis and responded to the other reviewers’ comments accordingly. 

Reviewer 3

Thanks for the submission and choosing older CALD women as the subject.

However, I have some observations:

1. The detail of the data collection procedure should be mentioned. We have elaborated on data collection per the previous reviewer comments.

2. What is the time of the survey? Time of survey has been added per previous reviewer comments.

3. Is it reliable to use online survey among older CALD women to collect data? We have added to limitations, “Recent research has concluded that web-surveys are feasible and a suitable alternative for surveying older people (Kelfve et al, 2020), however we acknowledge that by not offering paper-based or different language alternatives that this may bias results by excluding subsamples who may lack digital or English literacy. Data recruitment and collection methods were impacted by COVID-19 restrictions, limiting opportunity for distribution of materials via public meetings and snowballing, which will have implicated survey and interview response rates.”

4. Among 109 participants, only 25 patients agreed for the telephone interview and 10 of them were unavailable or could not be reached. This is the weakest part of this paper. 

5. Are there any significant differences of the risk perception and coping among the older CALD women and men? If yes, please mention those. And if not, why do the older men were excluded from this study? Please explain. This study is focused on women’s experiences. It is not a comparative study and men were not ‘excluded’. We already provided a justification for our study of CALD women based on literature cited which indicates they may be at heightened risk of poor coping. Accordingly, we have removed an erratum in the Methods, Research design, where we stated ‘subsample’ which is now corrected to “sample of older CALD women”.

---

## [Editor Report · Decision Letter 1]

6 Mar 2024

PONE-D-23-24279R1Manuscript: Older women experience with COVID-19 pandemic: A study of risk perception and coping among culturally and linguistically diverse population in South AustraliaPLOS ONE

Dear Dr. Hamiduzzaman,

Thank you for submitting your revised manuscript to PLOS ONE. After careful consideration, we feel that it has merit but does not fully meet PLOS ONE’s publication criteria as it currently stands. Therefore, we invite you to submit a revised version of the manuscript that addresses the points raised during the review process.

The detailed comment / required changes are provided below. 

Kind regards,

Ahsan Saleem, PharmD, PhD

Academic Editor

PLOS ONE

Journal Requirements:

Editor Comments:

Thank you for submitting your work to PLOS One. Based on the careful review of your revised submission, and taking into account the feedback from two independent reviewers and your responses to queries, we invite you to make another minor revision and resubmit your manuscript for consideration for publication.

1- Could you add an operational definition of Culturally and linguistically diverse (CALD) to the manuscript? Have a look at at how Australian Institute of Health and Welfare defines CALD: https://www.aihw.gov.au/getmedia/f3ba8e92-afb3-46d6-b64c-ebfc9c1f945d/aihw-aus-221-chapter-5-3.pdf.aspx

2- I didn't like the term Non-English speaking Europeans. Almost all the respondents were from Non-English speaking background so there is no need to make this distinction.

3- The abstract uploaded on the portal states that 67% women were non-English speaking CALD and they were still provided with English language questionnaire? Were interviews also conducted in English? Why?

4- I guess it would be better to say that the participants had somewhat English proficiency, but their spoken language at home was not English. Please fix this throughout the manuscript. If you add an operational definition of CALD, it would rectify this confusion.

5- Native-Australians term in confusing. Instead of writing CALD vs native-Australians, could you make it CALD vs. non-CALD throughout the manuscript. Native literally means 'born somewhere', so it can be confused with other people born in Australia, who could be anyone, may be First Nation Australians, Anglo-Australians or CALD Australians.

---

## [Author Response · Author response to Decision Letter 1]

10 Mar 2024

1- Could you add an operational definition of Culturally and linguistically diverse (CALD) to the manuscript? Have a look at at how Australian Institute of Health and Welfare defines CALD: https://www.aihw.gov.au/getmedia/f3ba8e92-afb3-46d6-b64c-ebfc9c1f945d/aihw-aus-221-chapter-5-3.pdf.aspx The operational definition of CALD has been added. 

2- I didn't like the term Non-English speaking Europeans. Almost all the respondents were from Non-English speaking background so there is no need to make this distinction. The manuscript has been reviewed to address your concern of using Non-English Speaking Europeans. 

3- The abstract uploaded on the portal states that 67% women were non-English speaking CALD and they were still provided with English language questionnaire? Were interviews also conducted in English? Why? The language used for both survey and interviews was English. This was made possible by the fact that the participants have had some level of English proficiency, even though their spoken language at home was not English. This is one of the limitations of this study. 

4- I guess it would be better to say that the participants had somewhat English proficiency, but their spoken language at home was not English. Please fix this throughout the manuscript. If you add an operational definition of CALD, it would rectify this confusion. An operational definition of CALD has been added to avoid confusion. 

5- Native-Australians term in confusing. Instead of writing CALD vs native-Australians, could you make it CALD vs. non-CALD throughout the manuscript. Native literally means 'born somewhere', so it can be confused with other people born in Australia, who could be anyone, may be First Nation Australians, Anglo-Australians or CALD Australians. Thank you for your suggestions. Changes have been made throughout the manuscript.

---

## [Editor Report · Decision Letter 2]

14 Mar 2024

Manuscript: Older women experience with COVID-19 pandemic: A study of risk perception and coping among culturally and linguistically diverse population in South Australia

PONE-D-23-24279R2

Dear Dr. Hamiduzzaman,

We’re pleased to inform you that your manuscript has been judged scientifically suitable for publication and will be formally accepted for publication once it meets all outstanding technical requirements.

Kind regards,

Ahsan Saleem, PharmD, PhD

Academic Editor

PLOS ONE

Additional Editor Comments:

Thank you for making the requested changes and resubmitting your manuscript for further consideration. We are pleased to inform you that your manuscript has been judged scientifically suitable for publication and will be formally accepted for publication once it complies with all outstanding technical requirements.

When you receive a decision of editorial accept, this will be your last opportunity to correct any errors in your manuscript. We have noted some grammatical errors in your manuscript and suggest having it reviewed by a language Editor or a native speaker. Please note PLOS ONE does not copyedit accepted manuscripts. Any typographical or grammatical errors should be corrected at this stage. Please check your files very carefully, because you will not be able to check or change anything after this point. 

Suggested changes:

P4. Line 4. Please fix the entire sentence ‘In Australia, generally it is used mean individuals who have cultural background different from the majority of Anglo-Celtic Australian culture.’P4. Line 7. Please change ‘population in this paper we use the term to refer to those people’ to just ‘the term used in this paper refers to people who were born in non-English speaking countries and who do not speak English as the main language at home.’P4. Line 24. What is SA in ‘CALD adults in SA during the pandemic’? Is that South Australia? Please write fully as it hasn’t been defined earlier in the text.P10. Line 4. Please fix ‘non-English speaking selfnominated CALD’ to 'Europeans' in the following sentence ‘67% (n=73) of them were non-English speaking selfnominated CALD’.Remove 'Manuscript' from your title 'Manuscript: Older women experience with COVID-19 pandemic: A study of risk perception and coping among culturally and linguistically diverse population in South Australia'

---

## [Editor Report · Acceptance letter]

20 Mar 2024

PONE-D-23-24279R2 

PLOS ONE

Dear Dr. Hamiduzzaman, 

I'm pleased to inform you that your manuscript has been deemed suitable for publication in PLOS ONE. Congratulations! Your manuscript is now being handed over to our production team.

Kind regards, 

on behalf of

Dr. Ahsan Saleem 

Academic Editor

PLOS ONE